# Patients’ Perspectives of Interprofessional Collaboration in Breast Cancer Unit

**DOI:** 10.3390/healthcare11030332

**Published:** 2023-01-23

**Authors:** Dea Anita Ariani Kurniasih, Elsa Pudji Setiawati, Ivan Surya Pradipta, Anas Subarnas

**Affiliations:** 1Department of Pharmacology and Clinical Pharmacy, Faculty of Pharmacy, Universitas Padjadjaran, Sumedang 45360, Indonesia; 2Doctoral Program of Pharmacy, Faculty of Pharmacy, Universitas Padjadjaran, Sumedang 45360, Indonesia; 3Pharmacy Study Program, Akademi Farmasi YPF, Bandung 40293, Indonesia; 4Department of Public Health, Faculty of Medicine, Universitas Padjadjaran, Sumedang 45360, Indonesia; 5Drug Utilization and Pharmacoepidemiology Research Group, Center of Excellence in Higher Education for Pharmaceutical Care Innovation, Universitas Padjadjaran, Sumedang 45360, Indonesia

**Keywords:** breast cancer patients, interprofessional collaboration, Indonesia, qualitative

## Abstract

Interprofessional teamwork provides significant benefits for patients. However, qualitative research on interprofessional collaboration in the breast cancer unit is uncommon. Therefore, a qualitative study was conducted to assess the perceptions of outpatient breast cancer patients regarding interprofessional collaboration in the breast care unit of an Indonesian referral center hospital. The teamwork involved in the interprofessional collaboration included breast cancer specialists, pharmacists, and nurses. In this study, in-depth interviews were performed with nine breast cancer outpatients. All interviews were audio recorded, transcribed verbatim, and analyzed using thematic analysis. The findings were divided into two categories to gather breast cancer patients’ viewpoints on interprofessional collaboration: (1) obstacle components to interprofessional collaboration: incompleteness of health personnel, no justification from health personnel, no knowledge of patients about health professionals, no involvement of patients in the therapy decision making; (2) enabling elements: patient-oriented, patient expectations, collaboration among healthcare personnel, patient participation in interprofessional collaboration, health personnel responsibilities, comprehensive hospital services. Respondents assumed interprofessional collaboration positively. However, several obstacles must be overcome to implement interprofessional collaboration in a breast care setting effectively. The research findings can be utilized to establish interprofessional collaborations aimed at improving quality healthcare in breast cancer units.

## 1. Introduction

In both high-income (HIC) and low- and middle-income countries (LMICs), breast cancer is the most prevalent malignant tumor among women [1]. Although breast cancer incidence is lower in LMICs than in HICs, mortality rates are higher in LMICs due to late-stage diagnosis and inadequate access to care [2]. Breast cancer is the primary cause of death in Indonesian women, according to GLOBOCAN 2020 data [3]. The diagnosis and treatment of cancer need multiple treatments, including surgery, systemic therapy (chemotherapy, immunotherapy, and endocrine therapy), and radiotherapy. As part of integrated, patient-centered care, a multidisciplinary team should administer these diagnostic and treatment approaches [4]. The multidisciplinary service team stresses taking a holistic approach, working collaboratively, and effectively communicating to ensure that patients’ diverse healthcare requirements are addressed and treated in an integrated manner [5]. Interprofessional teamwork is a fundamental skill for healthcare professionals (doctors, nurses, pharmacists, nutritionists, physiotherapists, and others) who provide patient care [6]. Collaboration requires effective cooperation [7,8,9] and communication [10,11,12]. Aiken and colleagues conducted a study demonstrating that a better work environment, which includes doctor-nurse relationships and interprofessional decision making, among other characteristics, is positively associated with increased patient satisfaction, quality of treatment, and safety [13]. Interprofessional collaboration has the potential to improve professional practice [14], the quality of life for patients [15,16], the satisfaction of health care providers [17], and job retention [18]. Interprofessional collaboration pleases the interest of not just researchers but also physicians, healthcare professionals, and policymakers, since it appears to be very promising not only for the quality of care but also for economic reasons [19]. Effective interprofessional collaboration does not always take place in hospital settings around the world, despite appeals for it [20]. Nevertheless, despite the existing body of knowledge regarding the potential positive outcomes of interprofessional collaboration, it remains a complex process to implement in the clinical setting [21] due to power imbalances between health care professionals [22], divergent understandings of interprofessional collaboration, or different professional backgrounds and interests [23,24]. When determining the existence of interprofessional collaboration aimed at improving patient clinical services, it is necessary for health care professionals to understand the factors that influence both the team’s strengths and weaknesses, so that it can be determined what steps should be taken to ensure the team’s sustainability and success [12].

A wide range of research has investigated health professionals’ perspectives regarding the factors that influence interprofessional collaboration [25,26,27]. On the other hand, patients’ opinions should also be considered an essential source for crucial and comprehensive data on the overall quality of care [28,29]. Patients’ perspectives appear to be underrepresented in the literature about interprofessional collaboration, even though we obtained some data on how this collaboration affects patient-centered care [30] and patients’ lack of understanding of interdisciplinary collaboration [31]. The relevance of the patient’s viewpoint in delivering high-quality, patient-centered care and enhancing the quality and efficacy of healthcare is becoming increasingly apparent on a global scale [32]. In a prior qualitative study, patients expected good care and collaboration from health professionals for improved health outcomes [33]. As suggested by current observational research on the efficacy of interdisciplinary team care [34], obtaining the patients’ opinion on interprofessional collaboration among healthcare professionals appears to be a promising strategy for delivering high-quality care [30]. In addition, the idea and concepts of interprofessional collaboration have long been investigated from the viewpoint of healthcare workers [35].

To date, limited qualitative research is present on interprofessional collaboration in the breast cancer units. Therefore, this qualitative study’s main aim is to assess patients’ perspectives concerning interprofessional collaboration in the breast care unit at a national referral center hospital in Indonesia.

## 2. Methods

### 2.1. Study Design

We adopted a qualitative research design to conduct in-depth interviews with nine breast cancer patients (*n* = 9) of diverse backgrounds, ages, and interprofessional collaboration experiences between December 2021 and July 2022. The inclusion criteria were women with breast cancer of grades II or III who had completed chemotherapy in cycles II, III, IV, V, or VI and were over 18. Patients with a Karnofsky Performance Status (KPS) Scale of at least 50 were eligible for this study. The KPS is a typical technique for assessing cancer patients’ ability to do everyday tasks; values range from 0 to 100. A higher score indicates that the patient is more capable of carrying out daily activities. The breast cancer patients were chosen for interviews and signed a consent form to approve their research participation. For the convenience of breast cancer patients, interviews were conducted at locations as determined by the patients, namely at the chemotherapy clinic, at the patient’s home, or via telephone or online means. Previous studies indicated that by establishing rapport with the subject, the person’s response would be enhanced [36]. Therefore, the interviewer began the conversation with general inquiries [37].

The in-depth interview session was divided into three phases. The first phase was the introduction of the interviewers to the participant. The second phase asked about participants’ experiences in getting breast cancer therapy, an overview of the collaboration among health workers, and participants’ experiences in a situation where the health workers explained the therapy to be given. In addition, participants were also asked about the barriers and facilitators while getting breast cancer therapy. In the third phase, participants were asked about their satisfaction and expectations with health services, the patient’s role in cancer therapy, and the follow-up of cancer services. Only the interviewee and interviewer were present at the site of an in-depth interview. Audio recordings were made of all of the interviews. Field notes were used to keep track of important facts during the interviews. All interviews were recorded with the agreement of the study participants and lasted between 10 and 40 min. There had been no previous contact between the researchers and the study subjects. Participants were informed that this study aimed to determine the current state of interprofessional collaboration in the breast cancer unit from the perspective of patients.

### 2.2. Context and Setting

The research was conducted at a tertiary hospital in West Java that also served as a breast cancer referral center. As a PhD student with a background in hospital pharmacy and an interest in interprofessional collaboration research, Dea Anita Ariani Kurniasih (D.A.A.K.) conducted the interviews. D.A.A.K. has received training in qualitative research methods, including in-depth interviews.

### 2.3. Information’s Trustworthiness and Credibility

Saturation was achieved when all the study participants with breast cancer were interviewed, and no new information was obtained from them. Recordings of in-depth interviews were made, and we voluntarily shared the transcription results with the participants so they might be corrected. Interviews were also reviewed against other sources, such as patient’s medical records, to ensure the information accuracy.

### 2.4. Data Analysis

The transcripts were examined using the method of thematic analysis [38]. Separately analyzing the transcripts and employing open coding, D.A.A.K. and Elsa Pudji Setiawati (E.P.S.) did an intermediate analysis, isolating meaningful phrases and concepts. Once a consensus was reached, the two researchers compared and analyzed their respective codes, after which they categorized the identified concepts into subcategories and more significant categories. Lastly, the researchers agreed on the final fundamental categories and subcategories presented in Figure 1.

### 2.5. Ethical Considerations

The project was permitted by the Universitas Padjadjaran Bandung Research Ethics Committee (number 098/UN6.KEP/EC/2021). All qualitative data are confidential and accessible to principal investigators. Pseudonyms were utilized in the transcriptions of the in-depth interviews and in any reports or publications arising from the study.

## 3. Results

Before conducting the research, an agreement was made between interviewers and participants that the interview would be conducted in two languages, Indonesian and Sundanese (West Java local dialect), because all participants could communicate in these two languages, both orally and in writing. Table 1 shows the participants’ characteristics from the in-depth interviews summarized according to gender, age, breast cancer stage, and occupation. All study participants were female, with the most significant proportion between the ages of 40 and 50, who had breast cancer of stages II and III. Among the participants, housewives were most frequently identified, as shown in Table 1.

### 3.1. Acceptability of Interprofessional Collaboration

Two characteristics of the acceptability of interprofessional collaboration emerged from the in-depth interviews. The two characteristics were obstacle components and enabling elements. Figure 1 depicts the categorization of themes and subthemes.

### 3.2. Obstacle Components

This study highlighted a variety of obstacle components to interprofessional collaboration, including the incompleteness of health personnel, no knowledge of patients about health personnel, and no involvement of patients in therapy decision making. The following are the comments from the breast cancer patients concerning the interprofessional collaboration and therapy while undergoing treatment at the breast cancer unit.

#### 3.2.1. Incompleteness of Healthcare Personnel


*‘During inpatient therapy, I felt that the nursing staff provided excellent care, but I only encountered nurses; why were there no doctors? If the expert physicians are unavailable, there may be other physicians who can attend.’*
—Participant F, 46 years old, private worker.


*‘Rarely doctors, if nurses are always there.’*
—Participant H, 56 years old, housewife.

The patient is concerned about the absence of the unit’s physician and a specialist healthcare professional. When an outpatient is set to undergo chemotherapy, the patient recognizes that the absence of one of the healthcare professionals raises concerns. With nurses who are constantly available to answer inquiries and respond promptly to patient complaints, patients feel at ease because they believe they can relay all complaints to these healthcare professionals.

#### 3.2.2. No Justification from Health Personnel


*‘The only thing that doesn’t have any explanation from the doctor is that it’s at least calming. For example, the doctor could calm the patient by giving an explanation.’*
—Participant C, 51 years old, housewife.


*‘Here, they only said later about the chemo. There was no explanation of my own illness until now. Even now, I still wonder, even though I read a lot about grades 1, 2, and 3, what about the stage and the immunohistochemistry results of positive ER I just read it because by reading it I guessed my illness without any explanation. Oh, I should like this. I must not be worried. It’s good. I think if a doctor says it, it will be cured, so I hope they make it more enthusiastic for the patient.’*
—Participant F, 46 years old, private worker.

Patients needed an explanation of the therapy planned to be administered by medical personnel. During the consultation with the physician, the patient understood the benefits of chemotherapy but had not received psychological support to add a sense of calm to carry out the therapy, nor had the explanation of the immunohistochemistry results previously been performed, so the patient remained curious about the relevance of the test.

#### 3.2.3. No Knowledge about Healthcare Professional


*‘I don’t know whether it’s a doctor or a nurse or who came there. So, I don’t know about that.’*
—Participant F, 46 years old, private worker.

The patient was unaware of which healthcare professional visited the breast cancer unit. Patients did not identify which health personnel came to the bed during chemotherapy since there is no relationship built on trust, equity, and a supportive atmosphere, which is crucial for patients who will undergo treatment according to the planned chemotherapy cycle.

#### 3.2.4. No Involvement in Decision-Making Process

‘*No one has yet been involved.’*—Participant F, 46 years old, private worker.

The patient felt that they were not involved in their breast cancer therapy decision making. Due to inadequate explanations from the resident doctor, during the discussion session at the beginning of the patient’s visit, the patient received limited knowledge, the patient’s treatment plan was not communicated, and patient participation in therapeutic decision making was also limited.

#### 3.2.5. Enabling Elements

The in-depth interview with patients also resulted in some enabling components that are important for interprofessional collaboration and improving patient care and safety. These involve patient-oriented, patient expectations, collaboration among healthcare personnel, patient participation in interprofessional collaboration, health personnel responsibilities, and comprehensive hospital services.

#### 3.2.6. Patient Oriented


*‘There was an explanation from the doctor. Indeed, there were three kinds of therapy, such as tumor removal, chemo, and oral medication up to 5 years.’*
—Participant C, 51 years old, housewife.

An explanation of therapy for breast cancer was obtained from the doctor during the first consultation meeting. Patients comprehend how treatment would be conducted initially, so they trust in the best-planned therapy for them.


*‘In my opinion, they are very good. They are also very professional in their profession, very friendly and serve the patient kindly so that the patient herself feels comfortable, safe, and calm during chemotherapy.’*
—Participant H, 56 years old, housewife.

The patient felt a relationship had been created with one of the health care professionals in the breast cancer unit, which increased trust and made the patient more at ease throughout therapy. Patients feel cared for because of the attitude of medical professionals and the treatment they receive.

#### 3.2.7. Patient Expectations


*‘The patient’s condition is especially important. If the patient’s condition is doing chemo, it must be healthy from leukocytes. The blood lab results must be good. For example, if there was a doctor who explained that later I would be like this, such as severe nausea and vomiting, for example, someone who kept on telling, for example, if there was a collaboration like that, maybe a doctor, nurse, and pharmacist, I, as a patient, will be like this. For example, medicine from the pharmacist can also be possible.’*
—Participant F, 46 years old, private worker.

Patients expect a comprehensive explanation of chemotherapy from health care professionals in the breast cancer unit, including side effects of therapy and how to manage them, how to interpret laboratory results for the requirements needed every time chemotherapy is administered, and what nutritional intake must be consumed to maintain laboratory results that meet the requirements, despite the fact that one of the side effects of chemotherapy is a reduction in blood test parameters.

#### 3.2.8. Collaboration among Healthcare Personnel


*‘The collaboration is also good, so we are also not stupid. The patients just seek treatment. They also give insight, so we also know. There is an explanation, so the cooperation is also good. Every time we go there, there is always new knowledge that I don’t know. What I don’t know becomes known.’*
—Participant B, 51 years old, private worker.


*‘I feel their planning is already good, so when the schedule has problems, the plan is dissolved and has to be planned again. It’s quite good interprofessional collaboration, whether with nurses I saw that day or with a specialist doctor.’*
—Participant I, 51 years old, civil government.

Patients witnessed how healthcare professionals work together to improve patient knowledge. Patients were offered the opportunity to experience interprofessional collaboration between nurses and physicians, in which these health professionals discussed modifications to patient care schedules.

#### 3.2.9. Patient Participation in Interprofessional Collaboration


*‘Sometimes the patient becomes the mediator of interprofessional or interprofessional communication. It is very important. I sometimes can’t answer questions like that, but maybe because my education background is in a health department and my environment is also health-related, I can learn the process from beginning to end, so when there is a change of service between nurses or between doctors, specialist doctors to general practitioners, specialist doctors and doctors’ residents, nurses and resident doctors, sometimes I become a mediator for communication.’*
—Participant I, 51 years old, civil government.

In the process of interprofessional collaboration, patients occasionally serve as a communication link between healthcare professionals. It will be easier to document if patients can read the laboratory result records that healthcare professionals request.

#### 3.2.10. Health Personnel Responsibilities


*‘The nurses were very good. They really served patients well and didn’t discriminate between patients. They worked hard to serve patients with their hearts and didn’t feel irritated or annoyed. Although there were many complaints from patients during chemotherapy, some were nauseous, they served them well.’*
—Participant H, 56 years old, housewife.

Health professionals, especially nurses, assisted patients with dedication and responsibility. During chemotherapy, patients said that nurses treated them with care. Especially when a patient is in distress as a result of medication side effects, nurses constantly pay attention and respond promptly.

#### 3.2.11. Comprehensive Hospital Services


*‘The staff’s reception at the hospital was warm. I am happy to think that I am accepted as part of the family. They really helped me, who didn’t know where to go before. They helped me deal with the hospital process.’*
—Participant B, 51 years old, private worker.

The chemotherapy scheduling process in the breast cancer unit is well organized, so patients know when to come in and submit laboratory results for chemotherapy. The patient was responsible for meeting administrative criteria, such as confirming that the insurance guarantee is still active. Because the process of mixing chemotherapy medications takes quite a long time in a separate area for synthesizing cytostatic drugs, patients were required to come in the morning to measure body weight and blood pressure and adjust the dose of chemotherapy drugs according to the regimen. This service and its dedicated health personnel have assisted patients. Comprehensive hospital facilities correspond to comprehensive patient care.

## 4. Discussion

According to the research, most patients had an interprofessional collaboration experience that provided a clear advantage to the patient. This is consistent with the hospital director’s policy on interdisciplinary cancer treatment practice [39]. In addition, earlier studies have supported this qualitative research that shows patients have engaged in interprofessional teamwork [12,19]. According to prior research, people with chronic diseases can participate in interprofessional collaboration as self-healing managers and representatives who contribute to therapeutic decision making [40,41]. In a different study, there were participants who did not want to be a part of the team since they trusted the healthcare professionals who treated them and acted simply as patients [30].

The outcomes of this experiment indicate the incompleteness of health personnel, in which the patients did not perceive the presence of a specialist health professional during chemotherapy. According to chemotherapy observation standard operating procedures, the nurse is responsible for monitoring the patient’s vital signs [42]. Although workers in health care in Indonesian health centers reported good interpersonal relationships, there may have been limited interprofessional contacts due to the hierarchical culture and lack of role interpretation among various professions [43]. According to the research of Van Dongen, it is crucial for all professionals participating in a patient’s care to present and apprise the patient’s current status as well as any recent updates, and they were especially delighted about the exchange of information between professionals from various fields and the patient [44]. Formal and social processes such as a team with low levels of conflict, supportive coworkers, open communication, and the team’s structure are crucial components of healthcare teams that develop environments for enhanced interprofessional collaboration [45]. Prior research demonstrated that nurses were more receptive to collaborating with other health professionals than other health professionals [46].

According to patients, the need for more explanation from health professionals regarding the patient’s condition, laboratory results, and planned therapy was another problem preventing interprofessional collaboration. According to Busari’s research, there is a substantial risk of losing domain-specific competence when procedures are not conducted or maintained in professional collaboration. A lack of clarity in patients’ treatment plans was viewed as a significant impediment [47]. A further impediment was the patient’s inability to recognize the healthcare practitioner, which might be due to their lack of familiarity. In prior research, it was found that patients would benefit from knowing healthcare professionals as persons with a particular background [25]. Professionals and patients recognize that a connection must be based on trust, equality, and a pleasant environment [44]. The next barrier is the patients’ lack of involvement in therapy decision making. In contrast, the patient or a family member was frequently involved in the previous study’s sessions, during which goals were examined while patient feedback was considered, and the patient was frequently asked how they thought the goals should be achieved or which areas demanded further attention [44]. Previous research indicated that the patient has a role in making the final decision about therapy in interprofessional collaboration, and that the patient must acquire knowledge, confidence, and skills beforehand in order to participate actively on the basis of trust, mutual respect, and competence [41].

Based on patient views, our research discovered that there are various benefits to the practice of interprofessional collaboration that can promote the success of breast cancer therapy. WHO research has also demonstrated that effective teamwork can boost patient happiness, provide greater access to healthcare, and enhance health outcomes [47]. Previous studies have proven that interprofessional collaboration and communication help improve the cancer journey [48]. Patient orientation is one of the things that patients go through before starting chemotherapy. The patient can prepare for everything if she understands what therapy is planned. This is consistent with earlier studies, indicating that health personnel’s attitudes toward patients are essential to demonstrate concern and assist with therapy [49]. Through this, the patient is able to declare, “The health care professionals are not conversing about me; they are speaking to me” [30]. According to other research, patients value a number of factors, including being treated seriously, receiving competent and compassionate care, being acknowledged as an individual in a particular circumstance, having appropriate time for discussion, and having access to treatment [50].

In treating breast cancer, patients may encounter interprofessional collaboration, wherein healthcare professionals pay attention to educating and planning chemotherapy for patients. This aspect is consistent with prior studies in which patients were involved in the practice so that they understood the treatment they would receive [30]. Another issue to consider is the patient’s understanding of the condition and the treatment plan. Patients believed they could not collaborate with their healthcare team until they were aware of their medical condition [30,51]. This study also reveals that patients are aware of the duty of healthcare professionals in patient care. This is consistent with O’Rourke’s research, which suggests using knowledge of one’s function and those of other professions to adequately assess and address patients’ medical needs and assist in improving population health [52]. According to Krug’s investigation, establishing a community of practice in an interprofessional team necessitates more time for adaptation processes and mindset shifts [53]. In the hospital, patients receiving breast cancer treatment should receive comprehensive care. This finding is congruent with The Supporting Room in Japan in Ishikawa (2008), which offers comprehensive services in a collaborative environment and is staffed by professionals from many professions [47].

According to our knowledge, this is the first study to evaluate the patient’s perspective on interprofessional teamwork within the breast care unit. This study’s findings may serve as a wake-up call for collaborative practice in the breast care unit, particularly in promoting patient-centeredness [30,54]. According to Thangarajoo, interprofessional learning requires self-assessment, attitude, and perspective among health professionals [55]. The findings can support the planning and implementation of additional quality improvement programs in the field of interprofessional care [30].

### Strength and Limitation

Even though this qualitative study investigated outpatient perceptions, no interviews with families or attendants who follow patients throughout treatment were conducted to determine the presence of interprofessional collaborative practices in the breast cancer unit. This can be accomplished with additional research. This study used several methodologies to strengthen validity and reliability, including triangulation and the member check method.

## 5. Conclusions

Depending on the patient’s background, patients’ impressions of interprofessional teamwork in the breast cancer unit vary. Patients had high hopes that they would be cared for by an interprofessional collaboration team that would help them comprehend the risks and benefits of the therapy. By understanding the risks and advantages, the patient can make knowledgeable choices and be well prepared.

## Figures and Tables

**Figure 1 healthcare-11-00332-f001:**
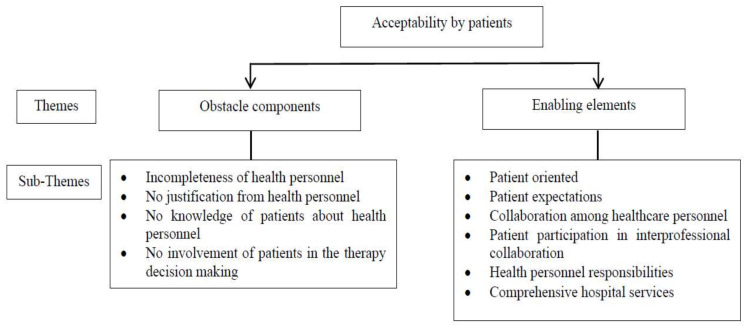
Acceptability by patients regarding interprofessional collaboration.

**Table 1 healthcare-11-00332-t001:** Characteristics of participants.

No	N	Age	BC Stage	Occupation
1	Respondent A	44	II	Housewife
2	Respondent B	51	II	Private worker
3	Respondent C	51	II	Housewife
4	Respondent D	49	III	Housewife
5	Respondent E	60	III	Housewife
6	Respondent F	46	II	Private worker
7	Respondent G	48	III	Private worker
8	Respondent H	56	III	Housewife
9	Respondent I	51	III	Civil government

BC: breast cancer.

## Data Availability

All audio cassettes and typed transcripts are stored in the author’s data repository. Due to ethical concerns, the datasets gathered and analyzed during this research are not available to the public; however, they are available from the respective author upon reasonable request.

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
