# Peer review of "Patients’ Perspectives of Interprofessional Collaboration in Breast Cancer Unit"

_healthcare, 2023, doi:10.3390/healthcare11030332_

Round 1
Reviewer 1 Report
The article of Kurniasih tries to evaluate the effect of interprofessional teamwork on the health of breast cancer patients. However, the introduction could be improved and the results and discussion are poorly written, not in-depth, and not commented on with other works of literature.
Reviewer 2 Report
Introduction: "Interprofessional teamwork is a fundamental skill for healthcare professionals (doctors, nurses, pharmacists, nutritionists, physiotherapists, and others) who provide patient care". Is it only the existence of an interprofessional teamwork fundamental? What about the quality of this teamwork, e.g. communication?
Methods: what does the Karnofsky Performance Scale represents? Please give a little more details to the reader (Line 71).
Results: why is gender presented and even collected in this survey? This is a specific survey regarding breast cancer in women.
Reviewer 3 Report
Thank you very much for offering this opportunity to review this manuscript. I enjoyed reading it. The purpose of this study is “to assess the perceptions of outpatient breast cancer patients regarding interprofessional collaboration in the breast care unit of an Indonesian referral Centre hospital.” This paper is generally well written, well-structured and concise. Findings of the study contribute to facilitating establishment interprofessional collaborations aimed at improving quality healthcare in breast cancer units. I have only one minor comment for the authors to consider:
Page 2, line 95: would it be better to change DAAK as the first author (DAAK)? The same as in line 106 for DAAK and EPS. Otherwise, it seems that all and sudden there comes an abbreviation.
